# Arsenic Distribution Assessment in a Residential Area Polluted with Mining Residues

**DOI:** 10.3390/ijerph16030375

**Published:** 2019-01-29

**Authors:** Carlos B. Manjarrez-Domínguez, Jesús A. Prieto-Amparán, M. Cecilia Valles-Aragón, M. Del Rosario Delgado-Caballero, M. Teresa Alarcón-Herrera, Myrna C. Nevarez-Rodríguez, Griselda Vázquez-Quintero, Cesar A. Berzoza-Gaytan

**Affiliations:** 1Facultad de Ciencias Agrotecnológicas, Universidad Autónoma de Chihuahua, Avenida Pascual Orozco s/n, Campus I, Chihuahua, Chihuahua 31200, México; manjarrez.carlos@gmail.com (C.B.M.-D.); mcnevarez@uach.mx (M.C.N.-R.); gquintero@uach.mx (G.V.-Q.); cberzoza@uach.mx (C.A.B.-G.); 2Facultad de Zootecnia y Ecología, Universidad Autónoma de Chihuahua, Periférico Francisco R. Almada Km 1, Chihuahua, Chihuahua 31453, México; jesus_prieto06@hotmail.com; 3Centro de Investigación en Materiales Avanzados S.C., Calle Cimav 110, Ejido Arroyo Seco, Durango, Durango 34147, Mexico; rosariodelgado369@gmail.com (M.D.R.D.-C.); teresa.alarcon@cimav.edu.mx (M.T.A.-H.)

**Keywords:** mining wastes, modeling, environmental problems, sustainable development, geospatial, geoestatistics, interpolation, GIS

## Abstract

Mining is a major source for metals and metalloids pollution, which could pose a risk for human health. In San Guillermo, Chihuahua, Mexico mining wastes are found adjacent to a residential area. A soil-surface sampling was performed, collecting 88 samples for arsenic determination by atomic absorption. Arsenic concentration data set was interpolated using the ArcGis models: inverse distance weighting (IDW), ordinary kriging (OK), and radial basis function (RBF). For method validation purposes, a set of the data was selected and two tests were performed (P1 and P2). In P1 the models were processed without the validation data; in P2 the validation data were removed one by one, models were processed every time that a data point was removed. An arsenic concentration range of 22.7 to 2190 mg/kg was reported. The 39% of data set was classified as contaminated soil and 61% as industrial land use. In P1 the method of interpolation with the lowest RMSE was RBF (0.80), the highest coefficient of E was RBF (46.25), and the highest *Ceff* value was with RBF (0.48). In P2 the method with the lowest RMSE was OK (0.76), the highest E value was 50.65 with OK, and the *Ceff* reported the highest value with OK (0.52). The high arsenic contamination in soil of the site indicates an abundant dispersion of this metalloid. Furthermore, the difference between the models was not very wide. The incorporation of more parameters would be of interest to observe the behavior of interpolation methods.

## 1. Introduction

The development of mining in Mexico has had a high impact on the environment. This activity, which has been the economic basis for the foundation of several communities, unfortunately it also generated a large amount of liquid, solid and gaseous wastes, mainly in form of sewage, gas, and slags [1]. The most common potentially toxic elements (PTE) derived from mining activities are lead (Pb), cadmium (Cd), zinc (Zn), arsenic (As), selenium (Se), and mercury (Hg) [2]. The PTE do not decompose through the processes of natural degradation, they have low mobility in soil, so they are accumulate over time [3]. The presence of metals in soil represents a risk for human health, since metal ions can be absorbed through inhalation of dust and food intake [4]. Previous studies have revealed that human exposure to high concentrations of some soluble heavy metals affects the central nervous system and may also disrupt the functioning of some organs [5,6]. Studies performed on dams in Chihuahua City reported high concentrations of heavy metals in fishes, an important bioindicator of their presence in the local abiotic environment and representing a health risk due to their ingestion [7,8].

Currently, mining is considered as one of the most important sources of contamination of heavy metals and metalloids [9]. Determining the spatial distribution of heavy metals could establish the basis for risk assessment in human health and support the development of environmental-urban management policies [10], however, there are constraints in the number of samples, for which the estimation of values in non-sampled areas (interpolation) becomes necessary [11]. Depending on the type of analyzed data, cost and accessibility of the site, it is determined how valuable the interpolation is [12,13,14]. Data interpolation offers the advantage of projecting maps or continuous surfaces, however, the amount of data in the studied area could limit its use [11]. Geostatistics is an efficient method for the study of the allocation of spatial characteristics of the soil and its spatial variation [12,13]. The use of deterministic and geostatistical techniques in the description of the distribution of metals and metalloids has been demonstrated by other researchers [14,15,16]. 

Robinson and Metternicht [17] used three different techniques, including kriging (geostatistical method), radial basis function (RBF), and inverse distance weighted (IDW), both deterministic, for surface estimation of salinity, acidity, and organic matter in soil. Pang et al. (2011) [18] reported that Ordinary Kriging (OK) is the most common type of Kriging in practice and provides a better estimate of soil properties. The use of interpolation methods (IDW, OK, and RBF) leads to the search for the most appropriate for the spatial estimation of metals in soil.

Due to this, the objective of this research was to compare the IDW, OK and RBF interpolation techniques using geographic information systems (GIS), to estimate and map the spatial distribution of As in the settlement Laderas de San Guillermo adjacent to a site with mining wastes in the municipality of Aquiles Serdán, Chihuahua, Mexico. Cross-validation will be applied to evaluate the best greatest fit com-pared to field data. The information derived from this study will be useful for the generation of urban-environmental policies focused on disease prevention by metals and metalloids. The mapping of the present research will be evidence to demonstrate the existence of the distribution of As in the residential area.

## 2. Materials and Methods 

### 2.1. Research Area

The studied area is located in San Guillermo, municipality of Aquiles Serdán, Chihuahua, Mexico, in the geographical coordinates among the parallels 28°27’ and 28°43’ north latitude; and the meridians 105°41’ and 106°00’ west longitude; altitude between 2100 and 2300 m. It borders to the north with Aldama, to the east with Julimes municipalities, to the south with Rosales and to the west with the municipality of Chihuahua (Figure 1). The climate of the area is classified as semiarid, its maximum temperature is 40 °C and its minimum −14 °C, with an average annual rainfall of 350 mm, and rainy season of 60 days average. The prevailing winds vary throughout the year. The east winds have duration of four months (June to October), while the west winds have a greater frequency (October to June), lasting from seven to eight months [19]. The area involves a housing zone located at 400 m of distance from deposits of mining wastes. The residues are found at open air, there it is fine material of an inactive extraction industry, which used the flotation method for mineral separation, where it was extracted mainly Pb and Zn. The production material used in the benefit plant was extracted from the mines of the area of Santa Eulalia, Aquiles Serdán, Chihuahua. These mines were outstanding for their production of gold, silver, lead, zinc, and copper, throughout the mining history in the region [20].

### 2.2. Soil Sampling

Sampling of surface soil, aimed for As quantification, was performed according to NMX-AA-132-SCFI-2006 [21]. Dust samples in the housing zone were collected according to the US Environmental Protection Agency AP-42 section 13.2.1 [22]. Sampling was realized within an area of 53 ha by a systematic technique based in a radial grid, with a spacing of 100 m and 15° of angle (Figure 2). A total of 88 samples were collected on spring in the residential area close to the mining wastes and surface soil in the neighborhood.

### 2.3. Arsenic Determination

The 88 samples were homogenized and sieved to a degree of fineness ≤75 μm. The homogenization of the samples was done by manual quartet. Acid digestion was made in a microwave MARSx CEM using the digestion method SW 846–3051 recommended by the manufacturer. Samples were prepared by placing 0.5 g of soil + 10 mL of nitric acid (HNO_3_), considering a blank. Once digested the samples were filtered and diluted up to 50 mL with distilled water.

Analytical As determination was performed by atomic absorption spectrometry (AAS) with a hydride generator in a GBC-brand AVANTA SIGMA (Waltham, MA, USA). Triplicate and control samples were taken. The As quantification limit was 0.005 mg/kg. The As concentrations were compared with the reference levels indicated in NOM-147-SEMARNAT/SSA1-2004 [23]. These were classified for residential use (less than 22 mg of As/kg of soil), industrial use (less than 260 mg/kg), and soil above the reference level [23], as shown in Table 1.

### 2.4. Spatial Distribution of Arsenic by Interpolation Methods

A layer of vector information of point type was created based in the As concentrations with software ArcGis 10.3^©^ (ESRI, Redlands, CA, USA). The interpolations were made through the extension Geoestatistical Analyst Wizard (ESRI, Redlands, CA, USA). The interpolation methods used were IDW, OK, and RBF.

The IDW method to estimate the concentration of nonsampled sites uses the existing values around the research area. The values of the closest observations have greater influence than those that are further away; this influence decreases with the distance [24]. Equation (1) for obtaining the estimated values with IDW is as follows: (1)$$Z\left(S_{0}\right)=\overset{N}{\underset{i=1}{\sum }}\lambda \times Z\left(S_{i}\right)$$
where: Z(*S*_0_) = value to be estimated in place *S*_0_, *N* = number of observations close to the place to be estimated, λ = weight assigned to each observation to be used, weight decreases with distance, Z(*S_i_*) = observed value of place *S_i_*.

The radial base function (RBF) estimates values using a mathematical function that minimizes the overall curvature of the surface, resulting in a smooth surface passing exactly through the entry points. The estimation of non-sample values shall be calculated by the following Equation (2) [25]: (2)$$Z\left(S_{0}\right)=\overset{n}{\underset{i=1}{\sum }}\omega \;\Phi \;\left(\left|\left|s_{i}{-s}_{0}\right|\right|\right){+\omega }_{n+1}$$
where Z(*S*_0_) is the value to be estimated in the place *S*_0_, Ф(r) is the radial base function, r = ||*s_i_*–*s*_0_||, is the Euclidean distance between the estimated location and each location *s_i_*, and ω*_i_*, *i* = 1, 2, … *_n +_*
_1_, are the weights to estimate.

The ordinary kriging (OK) method incorporates the statistical properties of the observations (spatial autocorrelation). OK employs the semivariogram (possible combinations between pairs of points) to express spatial continuity, this quantifies the weight of the correlation as a function of distance. The data closest to a known point have greater weight or influence in the interpolation [26]. The OK method is obtained based on Equation (3) [27]:(3)$$\overset{n}{\underset{i=1}{\sum }}\lambda_{i}\;Ɣ\;\left[d\;\left(s_{i}{\;,\;s}_{j}\right)\right]+m=\;Ɣ\left[d\;\left(s_{i}{\;,\;s}_{j}\right)\right]$$
where *n* is the number of observations, *m* is the Lagrange multiplier, used to minimize constraints, *λ_i_* is the weight assigned to each observation (the sum of all weights equals 1), *i*, *j* are observations, 0 is the point of estimation, *s* is the point of estimation (measured variable), and d(*s_i_*,*s_j_*) is the distance between *s_i_* and *s*_0_ from the semivariogram. The interpolation models used in this research have as common parameter the data neighborhood, necessary to perform spatial searches [28]. In the case of OK, the determination coefficients (R^2^_OK_) were compared by means of the GS+^©^ program, in order to determine which of them fitted better to the semivariogram generation.

For the surface calculation by degree of contamination, data was interpolated with the IDW, OK, and RBF methods. The interpolation was made with the extension Geoestatistical Analyst Wizard using the As concentrations. After the interpolation, the resulting map from the Geostatical Analyst was rasterized. The raster files were classified in accordance to Table 1, which describes the soil classification by level of As contamination. The classified raster files were converted in a shapefile with a UTM Zone 13 North reference system. Finally, the surfaces were obtained by the level of As contamination.

### 2.5. Normality Analysis

For the tests performance by interpolation methods with GIS, a statistical analysis of the As concentrations with the SPSS 20.0^©^ software (IBM, CHI, USA,) was performed. The Kolmogorov-Smirnov (KS) normality test was applied. The distribution of As concentrations in the dataset was not normal, thus the As dataset was logarithmically transformed As (ln) to obtain a normal distribution (Figure 3).

### 2.6. Validation and Interpolation Comparison

For the validation of interpolation methods, crossvalidation with As(ln) data was used. A set of the data were randomly selected using the SPSS 20.0^©^ program (test data and validation data) (Figure 4). Two cross-validation tests were performed. The first test (P1) consisted on running the interpolation models without the selected validation data (one run per interpolation method).

The second test (P2) consisted of removing the selected validation data one by one and performing the interpolation with the remaining data. This procedure was performed 26 times by each method. With this, the absolute value of the deviation between the value estimated by the interpolation for both tests and the experimental data was determined.

For the methods precision, was used the root mean square error (RMSE), the estimated predictive effectiveness (E), the Nash and Sutcliffe Efficiency Coefficient [29], known as *Ceff*, and the coefficient of determination (R^2^) between experimental and estimated data. The RMSE was estimated by Equation (4) [27,30]:(4)$$\mathrm{RMSE}\;=\frac{\sqrt{\sum_{i=1}^{n}{{(y}_{i}-{\hat{y}}_{i})}^{2}}}{n}$$
where *y_i_* is the observed value at point “*i*”, *ŷ_i_* is the estimated value at point “*i*”, and *n* is the number of used points.

The prediction effectiveness of each method (E) was determined through the following Equation (5) [31]: (5)$$E\;=\left(\;1-\{\sum_{i=1}^{n}{\left[z(x_{i})-\hat{z}(x_{i})\right]}^{2}/\sum_{i=1}^{n}{\left[z(x_{i})-\bar{z}\right]}^{2}\}\right)100$$
where z(*x_i_*) is the observed value at point “*i*”, $\hat{z}$(*x_i_*) is the estimated value at point “*i*”, *n* is the number of used points, $\bar{z}\;$ is the sample average. E equal to 100%, indicates a perfect prediction.

The coefficient *Ceff* was determined by the following equation: (6)$$Ceff\;=\;1-{\left(\frac{\mathrm{RMSE}}{\mathrm{SD}}\right)}^{2}\;$$
where RMSE is the root mean square error, and SD is the standard deviation [32].

The coefficient of determination (R^2^) was applied to determine the association between the experimental concentrations and the estimated data by the different methods (Equation (7)). The linear correlation between the experimental vs estimated values was done using the GLM procedure [33]:(7)$$R^{2}=1-\frac{\sum_{i=1}^{n}{{(y}_{i}-{\hat{y}}_{i})}^{2}}{\sum_{i=1}^{n}{{(y}_{i}-{\bar{y}}_{i})}^{2}}$$
where *y_i_* are the observed values, *ŷ_i_* are the estimated values, ȳ are the mean values, and *n* is the number of observations used.

## 3. Results

### 3.1. Arsenic Characterization

The As concentrations obtained from the soil samples reported a minimum value of 22.7 and a maximum value of 2190 mg/kg. The As data in soil indicated that 100% of the samples exceeded the reference level for residential land use (22 mg/kg). The 61% of the samples were within the range levels of industrial land use (260 mg/kg). Finally the 39% were classified as contaminated or severely contaminated soil (>260 mg/kg) (Table 2).

These concentrations demonstrated the dispersion of this contaminant to the inhabited zone, causing a high degree of exposure to As particles, and with this a high risk to public health for the inhabitants of the zone, since the As in soil and dust can be absorbed in human body through dermal contact, inhalation, and ingestion exposures [34].

### 3.2. Spatial Distribution of Arsenic

The As distribution in the studied area by the interpolation methods indicated a similar geographical trend, where high concentrations were found in the southern area. In other areas, As concentration was lower. The spatial distribution of As displayed differences due to the mode in which the data were statistically grouped by each interpolator [27].

The As concentration decreased as the zone was farther from the mining wastes, however, a part of the inhabited area was covered by high As concentrations (>260 mg/kg, above reference level for industrial land use). The other part of the inhabited area was covered by As concentrations from 22 to 260 mg/kg, which ones are above reference level for residential/agricultural use. Possibly the As direction is associated with the land topography where the elevation varied from 1550 to 1510 masl (Figure 1), with a slope of 17.5%. There is also a barrier formed by hills in the east, coupled with the 60 days of average rainfall, contributing to its dispersion, which agrees with the mentioned by Delgado-Caballero (2018) [35]. The spatial distribution of As by the three different interpolators visually tends to be similar. The IDW method has a greater tendency to form hotspots, this tendency is common where surrounding data values are closer to the estimated value [36] (Figure 5).

### 3.3. Surface by Arsenic Range

Considering the predominant ranges of As concentrations in the experimental sampling (Figure 1 and Figure 2), in the classification of industrial land use (22–260 mg As/kg) the IDW method estimated an area of 21.39 ha, OK an area of 21.89 ha and RBF 23.37 ha. In the contaminated soil range (260–1000 mg As/kg) IDW estimated an area of 21.44 ha, OK 20.81 ha and RBF 19.06 ha. In the severely contaminated soil range (>1000 mg As/kg) the IDW method reported a surface area of 1.88 ha, OK of 2.04 and RBF of 2.3 ha. The estimated data are presented in Figure 6 and Table 3.

The IDW method reported a minimum estimated value of As of 15.3 and a maximum of 2183 mg/kg; OK a minimum value of 36.0 and a maximum of 1989 mg/kg; and RBF a minimum value of 17.0 and maximum of 2.166 mg/kg. The above display a great similarity between the minimum and maximum As values in the IDW and RBF methods in comparison to the experimental values. The OK values were distant from the experimental data. This can be explained by the nature of the interpolators. The IDW and RBF methods are exact interpolators, which generate the estimated surface by adjusting the values to the real data, marking abrupt changes in the estimated surface [37]. The OK eliminates the highest and lowest values to obtain a smaller error in the estimation, which causes a smoothing in the estimated surface [38]. This explains the discrepancy of the maximum and minimum values of this model with the experimental ones, and justifies the discretion in the calculation of the surface compared with the other two methods.

### 3.4. Validation and Interpolators Comparison

The mean and the variation for the estimated values of As(ln) in P1 and P2 by interpolation methods are presented in Table 3. The descriptive statistics demonstrated that for both tests, P1 and P2, the lowest variance corresponded to IDW (0.33 and 0.35, respectively). The mean of the experimental data was 5.31, in P1 the method that approached the mean value was OK (5.48), and in P2 the method with the closest value to the mean was RBF (5.48). In P1 the amplitude between the minimum and maximum values were lower for IDW (4.44–6.65), followed by RBF and OK. The amplitude at P2 was lower for IDW (4.38–6.62), followed by RBF and, finally, OK (Figure 7).

The precision obtained by RMSE, E (%) and *Ceff* for the data of the interpolation methods are presented in Table 4. In P1 the method with lowest RMSE was RBF (0.800), followed by IDW (0.803). The interpolator with the highest coefficient of E was RBF (46.25), followed by IDW (45.5). The highest *Ceff* value was with RBF (0.48), then IDW (0.47). In P2 the interpolation method with the lowest RMSE was OK (0.76), followed by RBF (0.78). The highest E value was 50.65 with OK, followed by 48.19 with RBF. The *Ceff* reported the highest value with OK (0.52), followed by RBF (0.50). This result shows the difference between both validation tests. In P1 the most accurate method was RBF while in P2 the most accurate method was OK.

In Figure 8 and Figure 9, the data distribution is graphically presented by comparing the experimental As(ln) values with those estimated by the interpolation methods. In P1 the highest coefficient of determination (R^2^) was with IDW (0.538), followed by RBF (0.533). For P2 OK presented the highest value of R^2^ (0.54), followed by RBF (0.528). According to the data obtained the IDW and RBF methods are very similar in precision in P1 showing high values of E(%) and R^2^. However, in P2, OK was the most accurate in agreeing with the highest values of E(%) and R^2^ (Table 5). High R^2^ and low RMSE values indicate a good agreement between observed and estimated concentrations of As at the different sampling points (Table 5).

The statistical analysis of P1 and P2 shows that the smallest variation of the estimated data was with IDW and OK the one with the highest variance. The precision parameter for interpolation methods indicated that in P1 RBF was the most accurate, while in P2 was OK. The correlation coefficients between the observed values and those estimated at P1 were higher with IDW. In P2, OK exposed a significant correlation coefficient (*p* < 0.0001).

Observing the estimates of each of the models and tests, good correlations were obtained between models of the same test (>0.95), however comparing the models between correlation tests were low (<0.60) (Table 6). Those results suggest, that behavior between tests is different and the difference between models is the same.

### 3.5. Error Mapping

In the error maps, the areas in red indicate a greater degree of error between the experimental and estimated data. Zones in blue indicate an intermediate degree of error. Zones with yellow indicate a lower error rate, which represents surfaces with values close to the real ones. The zones with a high error rate for the three interpolation models concur in the northwestern part, at the boundaries of the studied area and in close proximity to a topographic transitional zone. This transition zone corresponds to the boundary between the residential area and the hills beginning; moreover it is the further zone of the contamination source. Thus, this area could be subject to greater variability.

For P1 the interpolation of the errors between the experimental values and estimated by each method are presented in Figure 10. The OK and RBF methods present great similarity in the yellow surface (low error degree). IDW modeled in the northeastern part of the studied area a zone in yellow tonality, revealing uncertainty between the experimental data and the As estimated. This showed that although the IDW method had the lowest variance and the highest value of R^2^. The values estimation presents a greater error in the spatial distribution. For RBF homogeneous surfaces are observed in the central pair in yellow tonality, which reaffirms their accuracy in P1.

The interpolation errors in P2 are presented in Figure 11. A zone in the southern part generated by the IDW method presents a hotspot effect with an intermediate error rate. In the northwestern part, the three interpolation methods show the same zone with high error. It is possible to see that the OK method shows different surfaces in yellow tonality, indicating a smaller error between the experimental and estimated values. Following, RBF presents several zones with a low error rate. Finally, in IDW, zones with an intermediate error degree are observed, making it less reliable for P2. However, this test had a greater error than the P1 test.

## 4. Discussion

Soil and dust are important components in the housing environment and, therefore, have the most significance in the effects of human health. Exposure to As could cause harmful effects to human health, such as skin lesions, cardiovascular diseases, and metabolic disorders [34]. The spatial distribution suggests the control of high concentrations of the metalloid (up to 2190 mg As/kg) in the neighborhood of the deposit of mining residues, marking this zone as the area with the highest concentration of As. The southern zone of the housing complex also has high concentrations of As (from 22 to 1000 mg As/kg) indicating that the slope and direction of the wind is strongly associated with the distribution [39]. The lowest concentration of As was presented in the northern part of the housing complex, which match with the assumption related to be the farthest from the waste zone, however, it is worrying that these As concentrations exceed the reference levels for land use. The As dispersion is primarily caused by rain runoff due to slopes and wind transport in the direction from south to north [35].

Descriptive statistics expressed that in P1, the model with the best evaluation indicators was IDW, with the smallest variance, good precision and the highest correlation between estimated and observed values (*p* < 0.0001). In this test, RBF the model with more precision showed values very close to those of IDW in variance and correlation. In P1, KO was not the best in the precision parameters of RMSE, E (%), *Ceff*, and R^2^, however, the spatial distribution of the error shows surfaces in yellow very similar to RBF. In P2 the model with the highest precision and correlation was OK, nevertheless it was the one that presented greater variance; IDW presented a small variance and high precision and correlation values.

In general, with correlation tests, a high correlation was determined between models per test, however, a different behavior was observed between tests, which is according to how the experimental data are removed to run the model. Additionally, IDW and RBF are easier to use because they require fewer input parameters. In contrast, OK is more difficult to use. The typical OK includes several stages, such as statistical testing, data transformation, spatial structural analysis, and semivariogram function [40].

The error maps showed a high degree of error, on the northwest side (lowest As values). This could be due to the fact that in the zone with high concentrations of As can be present in a punctual way or the edge effect that affected the three methods of interpolation. The difference between tests (P1–P2) shows that OK best estimates the As(ln) values with less data (P2), in comparison to IDW and RBF. A higher data density can explain the superiority of RBF and IDW in P1. This shows that OK is better interpolator with a lower density of randomly distributed data, therefore, it is of greater use for the data simulation when the sampling and analysis is costly in the whole area, such is the case of As.

Bhunia et al. (2016) [10] compared interpolators for organic carbon in soil using IDW and RBF, in such research they found that the IDW method was better in regards to RBF among others. In a research by Fortis-Hernández et al. (2010) [41] about the dispersion of nitrate and ammonium in soil, the IDW method turned out to be the most significant, followed by RBF in comparison to OK. Mueller et al. (2001) [42] compared IDW and OK in the analysis of pH, P, K, Ca, and Mg, concluding that IDW was a better option than OK, in data lacking spatial structure. Robinson and Metternicht [17] compared the IDW, OK, and splines methods in the estimation of subsoil pH, soil pH, electrical conductivity, and organic matter. They found that IDW was better at interpolating subsoil pH, OK was better interpolator in surface soil for pH, while organic matter and electrical conductivity were better interpolated with splines. These comparisons with other types of soil elements show the great variability in the accuracy of interpolation methods. Yan et al. (2015) [43] tested the performance of the Kriging interpolator, finding that the mine activity had no influence in the contiguous zones. Chaoyang et al. (2009) [44] used the IDW and kriging interpolators and Fu and Wei [45] the kriging interpolator finding that high concentrations of heavy metals come from mining settlements, which is consistent with our findings.

The efficiency of pollution assessment depends on the accurate mapping of metals and metalloids. The factors that affect accuracy are the number of samples, the distance between the sampling points and the sampling method [46]. Generally, higher sampling density would produce more accurate contamination mapping of heavy metals [42]. However, due to the cost and time of the sampling as well as the analysis cost of the samples, a high sampling density is impractical [40].

With the error mapping, although the results showed few differences, it was possible to show the performance of the interpolation methods. This highlights the importance of verifying the results in a descriptive and spatial way to know the performance of the methods of interpolation [47]. In comparison with other studies [43,48,49,50], only the result of the spatial distribution of the variables under study is mapped, however, the mapping of uncertainty is essential to know the viability of the application of interpolation methods. Before proposing risk strategies it is important that the error mapping be verified as subsequent efforts may be poorly focused on the space and surface.

The increase in population, the demand for residential buildings [51], and the introduction of industries in areas with soils contaminated by mining waste pose a major health risk. This study tests the proximity and movement of arsenic in the human settlements of Aquiles Serdán. The interpolation maps are the basis for making decisions to minimize the health risk to the population.

Due to this, it is recommended to continue working with interpolation methods at these scales, for the management of the urban environment versus the As and other metals contamination. It is important to continue working with interpolation methods, nevertheless for the tests revealed in this research it is recommended to work at the submethod level of the interpolator and to verify the existence of significant differences between methods

## 5. Conclusions

The high environmental contamination of As present in soil of the studied housing area indicates a high dispersion of this metalloid presented in the mining wastes of the research area. The largest concentration of As is located in the southern part of the housing area, the one closest to the mining wastes. Waste deposits constitute a high risk for public health that must be addressed immediately, because if the contaminants dispersion in the area is not minimized, the health risk of the exposed population remains latent as the main routes of exposure: Inhalation of fine particles of airborne mining wastes, and particles suspended by vehicular traffic, as well as contaminated soil and dust intake.

Clear understanding of the distribution, extent and direction of As is the key in assessing the dispersion of such contaminant. As dispersion with the different interpolation methods IDW, OK, RBF was represented with similarities between them. The three methods enclosed the same area with higher As concentration and had minimal variation in the other classifications. Statistic data demonstrated that IDW was more precise in P1, and KO in the P2. Thus, the accuracy of the models could differ depending of the way of run the data. Although, differences between the models were not wide. The incorporation of more parameters, such as concentration of other metals, topography, soil type, and geology, among others, would be of interest to observe the behavior of interpolation methods.

## Figures and Tables

**Figure 1 ijerph-16-00375-f001:**
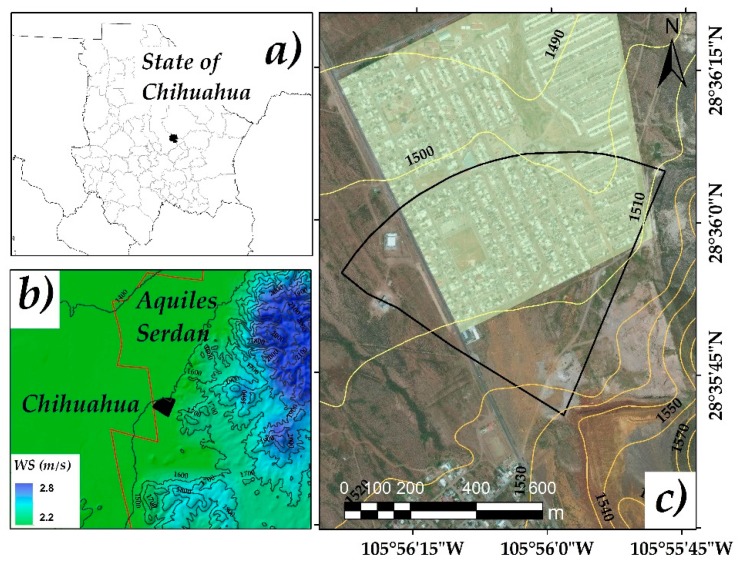
Study area location. (**a**) State of Chihuahua, (**b**) wind speed in the Municipality limits of Aquiles Serdán and Chihuahua, (**c**) research area in San Guillermo. Study area (**□**), human settlements (■), WS = wind speed, contour lines (~), and municipality limits (-).

**Figure 2 ijerph-16-00375-f002:**
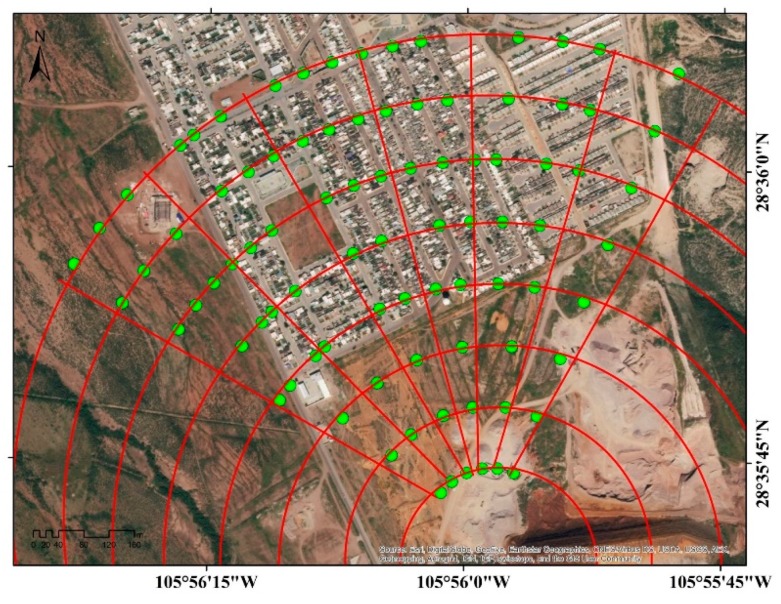
Design of radial sampling (*n* = 88). Design of radial (**—**). Samples (●).

**Figure 3 ijerph-16-00375-f003:**
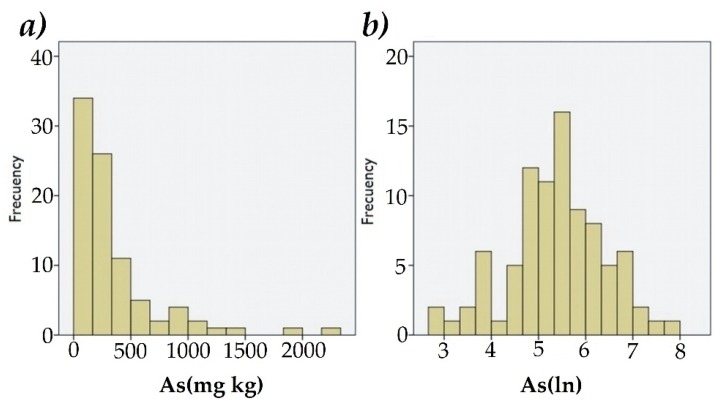
Frequency of As concentration: no logarithmic transformation (**a**), with logarithmic transformation (**b**). As = arsenic, As(ln) = arsenic values logarithmically transformed.

**Figure 4 ijerph-16-00375-f004:**
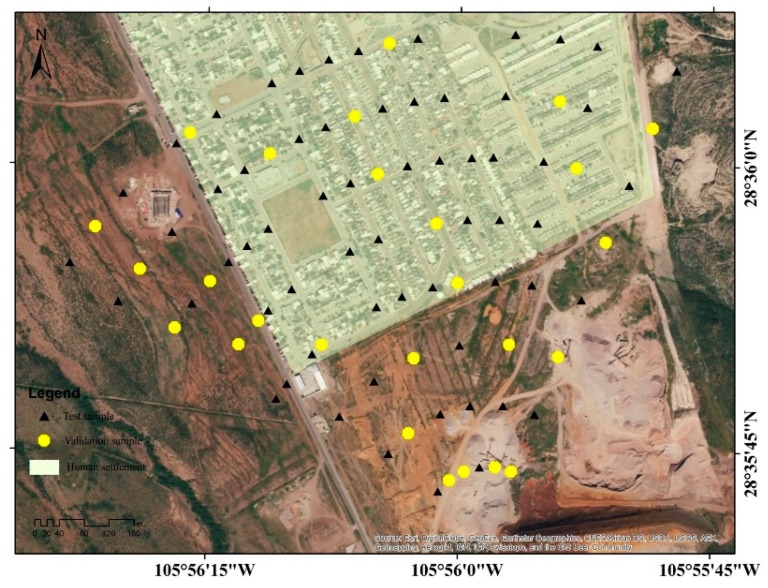
Test and validation points of the interpolations in P1 and P2. ▲ = test sample, ● = test validation.

**Figure 5 ijerph-16-00375-f005:**
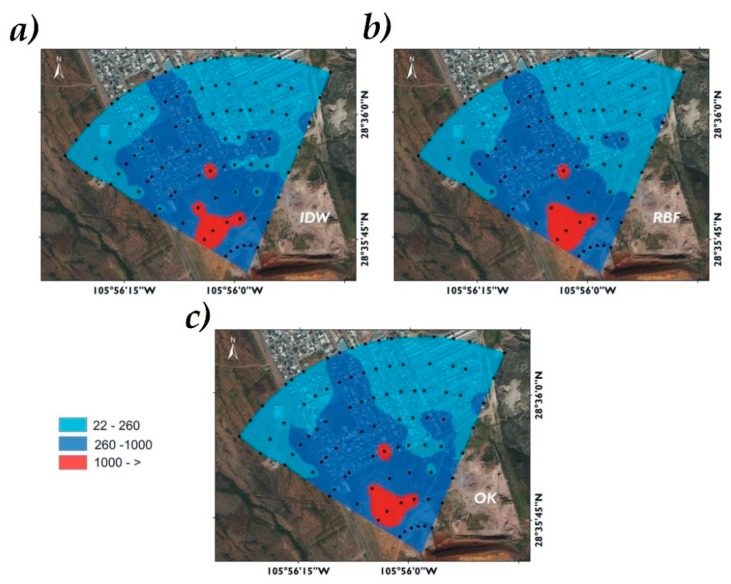
Spatial distribution of As in San Guillermo obtained by interpolation methods: IDW (**a**), RBF (**b**), and OK (**c**). Samples (●), Industrial use (■), Contaminated soil (■), Severely contaminated soil (■).

**Figure 6 ijerph-16-00375-f006:**
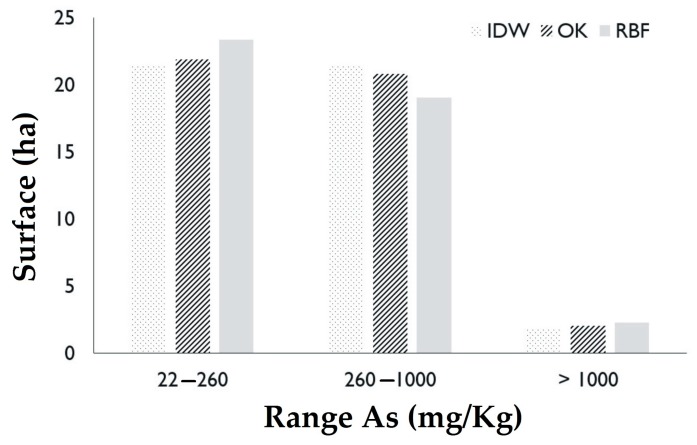
Surface by concentration range of As.

**Figure 7 ijerph-16-00375-f007:**
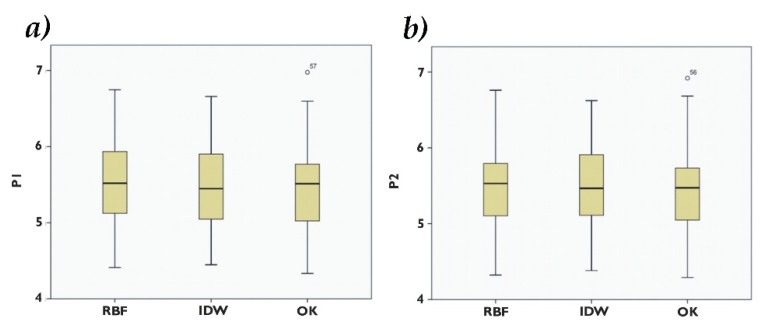
Box diagram for values of P1 (**a**) P2 (**b**). IDW = inverse distance weight, OK = ordinary kriging, RBF = radial basis function, P1 = Test 1, P2 = Test 2.

**Figure 8 ijerph-16-00375-f008:**
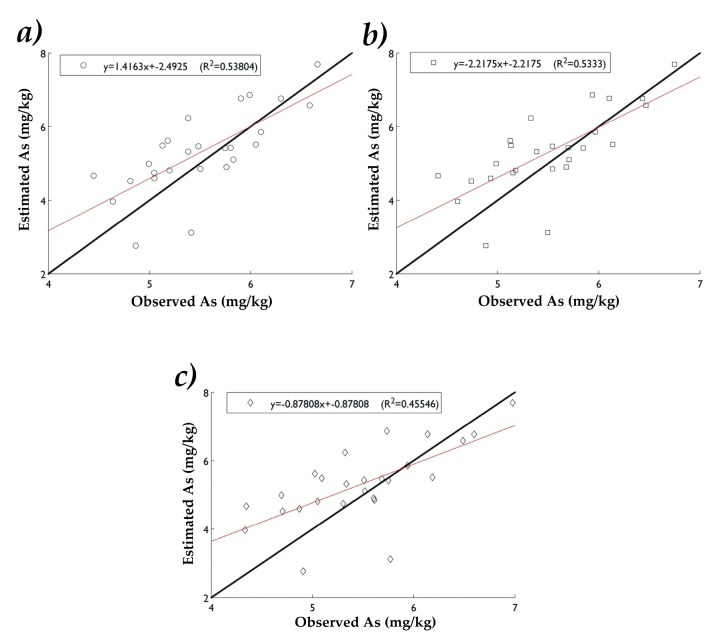
Correlation between experimental and estimated values of As(ln) in P1 by (**a**) IDW, (**b**) OK and (**c**) RBF.

**Figure 9 ijerph-16-00375-f009:**
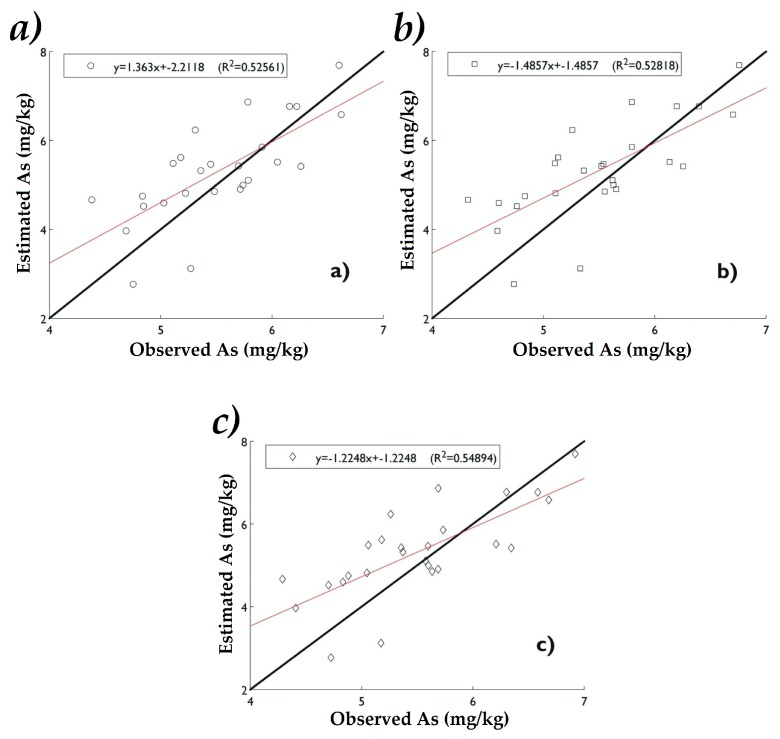
Correlation between experimental and estimated values of As(ln) in P2 by (**a**) IDW, (**b**) OK, and (**c**) RBF.

**Figure 10 ijerph-16-00375-f010:**
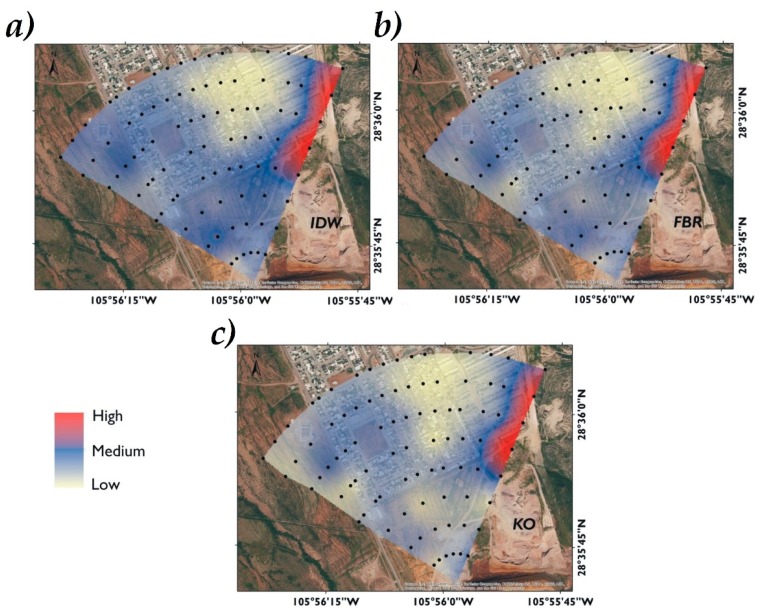
Error maps in P1 between experimental values and those estimated by interpolation models. IDW (**a**), RBF (**b**), and OK (**c**). Samples (●), errors between the experimental values and estimated values: high (■), medium (■), low (■).

**Figure 11 ijerph-16-00375-f011:**
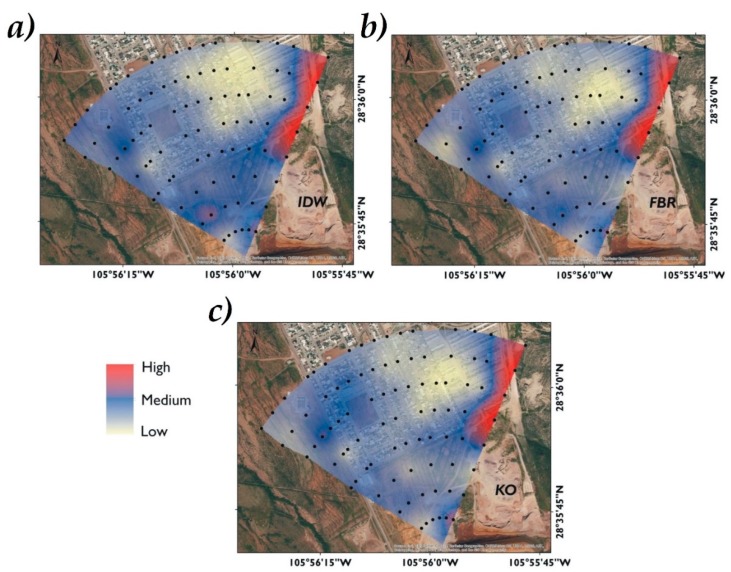
Error maps in P2 between experimental values and those estimated by interpolation models. IDW (**a**), RBF (**b**), and OK (**c**). Samples (●), errors between the experimental values and estimated values: high (■), medium (■), low (■).

**Table 1 ijerph-16-00375-t001:** Soil classification by arsenic values according to NOM-147-SEMARNAT/SSA1-2004.

Soil Classification	Range (mg As/kg)
Residential use	0–22
Industrial use	22–260
Contaminated soil	260–1000
Severely contaminated soil	1000–>

**Table 2 ijerph-16-00375-t002:** Soil classification by arsenic concentrations.

Classification	As Range (mg/kg)	As Mean (mg/kg)	Standard Deviation	Percentage of Samples
Residential use	0–22	-	-	0
Industrial use	22–260	122	71	61
Contaminated soil	260–1000	468	208	31
Severely contaminated soil	>1000	1421	428	8

**Table 3 ijerph-16-00375-t003:** Statistical data estimated by interpolation methods for P1 and P2.

Comparison Parameters	As(ln) Experimental	P1			P2		
		IDW	OK	RBF	IDW	OK	RBF
**Minimum value**	2.76	4.44	4.33	4.41	4.38	4.28	4.32
**Mean**	5.31	5.5	5.48	5.5	5.51	5.49	5.48
**Variance**	1.2	0.33	0.44	0.35	0.35	0.48	0.42
**Maximum Value**	7.7	6.65	6.97	6.74	6.62	6.91	6.75
**Typical error**	0.21	0.11	0.13	0.11	0.11	0.13	0.12

As(ln) = arsenic values logarithmically transformed, IDW = inverse distance weight, OK = ordinary kriging, RBF = radial basis function, P1 = Test 1, P2 = Test 2.

**Table 4 ijerph-16-00375-t004:** Accuracy of interpolation methods IDW, OK, and RBF.

Comparison Prameters	P1				P2	
	IDW	OK	RBF	IDW	OK	RBF
**RMSE**	0.803	0.82	0.800	0.80	0.76	0.78
**E (%)**	45.82	42.50	46.25	45.18	50.65	48.19
***Ceff***	0.47	0.44	0.48	0.47	0.52	0.50

IDW = inverse distance weight, OK = ordinary kriging, RBF = radial basis function, P1 = Test 1, P2 = Test 2, RMSE = root mean square error, E = estimated predictive effectiveness.

**Table 5 ijerph-16-00375-t005:** Accuracy of interpolation methods IDW, OK, and RBF through R^2^ for P1 and P2.

Test	Method	*n* ^†^	R^2^
P1	IDW	26	0.53 **
	RBF	26	0.53 **
	OK	26	0.45 *
P2	IDW	26	0.52 **
	RBF	26	0.52 **
	OK	26	0.54 **

*n***^†^** = test data, IDW = inverse distance weight, OK = ordinary kriging, RBF= radial basis function, P1 = Test 1, P2 = Test 2, * significant *p* < 0.05, ** highly significant *p* < 0.0001.

**Table 6 ijerph-16-00375-t006:** Pearson correlation coefficients, *n*
**^†^** = 26.

	IDWP1	RBFP1	OKP1	IDWP2	RBFP2	OKP2
**IDWP1**	1					
**RBFP1**	0.98 **	1				
**OK P1**	0.95 **	0.98 **	1			
**IDWP2**	0.51 *	0.51 *	0.44 *	1		
**RBFP2**	0.50 *	0.48 *	0.39 *	0.98 **	1	
**OKP2**	0.44 *	0.43 *	0.34	0.97 **	0.99 **	1

IDW = inverse distance weight, OK = ordinary kriging, RBF = radial basis function, P1 = Test 1, P2 = Test 2. * = Significant *p* > 0.05, ** = Highly significant *p* > 0.0001.

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
