# Peer review of "Arsenic Distribution Assessment in a Residential Area Polluted with Mining Residues"

_ijerph, 2019, doi:10.3390/ijerph16030375_

Round 1
Reviewer 1 Report
In general it is an interesting work, however not very novel.
I comment that for the studies of risk to human and ecological health the models of isoconcentration of As and metals do not have as much weight as they mention it, since although it is true that they are fundamental to define the risk zones, they are not for the decision making in materio of risk to human and ecological health. It is only one component of the many that exist to define risk (eg bioassays, risk estimation, biomarkers of exposure and effect, integration of evidence, intervention as risk communication), therefore, the mapping of concentrations is just one more evidence.
An example of which isoconcentration maps are not fundamental to determine if there is a risk is that there are very high concentrations of As and other elements in soil and dust, but that this is not bioavailable, that is why it is essential to know the characteristics physicochemical -mainly the pH- to know the bioavailability of the elements for the human and the biota of the region.
Author Response
Comments are in the attached document

Reviewer 2 Report
The manuscript entitled “Arsenic distribution assessment in a residential area polluted with mining residues” by Manjarrez-Domínguez et al. presents an interesting work on the use of different interpolation methods to estimate the As distribution in an area impacted by mining wastes. Nevertheless, some additions and corrections are necessary before accepting this manuscript for publication. Revision should be focused on the following points:
Abstract
· On page 1, in line 26, please, replace the number “2,189.9” by the number “2,190”.
Introduction
· On page 1, in line 42, please, correct the symbol of selenium (change S by Se).
Materials and methods
· On page 3, in line 91, the following comment is addressed “…being mainly the minerals of interest Pb and Zn”. Nevertheless, Pb and Zn are not minerals, but elements. Please, replace Pb and Zn for the minerals containing these elements in the studied area.
· Please, remove the foot note from Table 1.
Results and Discussion
· On page 6, in line 192, please, replace the number “2,189.9” by the number “2,190”.
· Please, remove the foot note from Table 2.
· On page 8, in line 224, please change the number “15.31” and “2,183.00” by the numbers “15.3” and “2,183”, respectively.
· On page 8, in line 225, please change the number “36.01” and “1,988.70” by the numbers “36.0” and “1,989”, respectively.
· On page 8, in line 226, please change the number “16.99” and “2,166.38” by the numbers “17.0” and “2,166”, respectively.
· Please, enlarge the size of the legends and the titles of x and y axes appearing in Figures 8 and 9 in order to make them readable.
Discussion
· On page 13, in line 230, please, replace the number “2,189.9” by the number “2,1890”.
Results and Discussion / Discussion
· Section 3 is entitled “Results and Discussion” and section 4 is entitled “Discussion”. Please, leave only one section entitled “Results and Discussion” or two sections entitled “Results” and “Discussion”, and reorganize accordingly.
Author Response
Comments are in the attached document

Reviewer 3 Report
This research deals with the assessment of As concentration in an adjacent zone to mining wastes interpolating experimental data using three ArcGis statistical models to obtain maps of As concentrations. The study is interesting and methodology is very useful for policy makers, in general the experimental methods are well with enough samples. Nevertheless some issues should be addressed before publication.
Materials and Methods are well explained, but it is important to specify when were taken the samples (season), since samples could be different depending on meteorological conditions. It is not mentioned if analysis were done by duplicate or triplicate, as well as quantification limits. In general quality assurance method is missing.
My major concern is related with the absence of topographic details of the zone, as well as the map wind. That is important since as the authors have been demonstrated not only the proximity of mining deposits is important in the presence of high As concentrations.
Those data must be included since authors claimed in Page 7 line 209 “it is was evident that the As dispersion and direction could be associated to the territorial characteristics of precipitation and topography” . It is not evident, since topography and precipitation were not explained”. On the other hand in Page 2 Line 84 it was mentioned that prevailing winds come from the southwest, but observing Figure 5 it seems that comes from south east. In addition authors say that results are in agreement with a study that cannot be consulted by internet since is a thesis wrote in Spanish. Then authors should include which are the results of Delgado-Caballero and the compare them.
Figure 5 is really nice. With these results authorities and policy makers could take some actions, but a detailed explanation of why highest As concentrations are in the red sites is missing. This explanation should include the run off, general topography and wind direction to understand the situation.
Table 3 is unnecessary since all data are in Figure 5.
Figures 8 and 9 should be improved.
Related to Figure 11. It is not clear in Page 11 Line 303 “Zones in the southern part generated by the IDW method presents the hotspot effect with a high error rate”. It seems to me that the high error rate is in the northwestern part”. Please clarify.
Page 11, Line 294 authors mentioned “a high error rate for the three interpolation models concur at the boundaries of the studied area and in close proximity to a topographic transitional zone”. It is necessary an explanation of the meaning of topographic transitional zone.
Discussion
Page 13 Line 323-324. Again authors claim that “the slope and direction of the wind is strongly associated with the distribution” when they have not related the wind direction and topography. The same 327-328 “dispersion of As is primarily caused by rain runoff due to slopes and wind transport in the direction of prevailing winds in the area (from South to North and from East to West” this is contradictory with prevailing wind mentioned in Page 2. The reference 35 as was mentioned before is not useful since cannot be consulted. And it is better to obrain the data from meteorological sources.
The sentence in page 13, Line 370 is incomplete.
“However, due to the cost and time of the sampling as well as the analysis cost of the samples, a high sampling density is [40]”
Conclusions
Authors say “This research explored the potential of, whereby the spatial distribution of As in the area was shown with interpolation methods, although the difference between the models was not wide”
This is not a conclusion, it is a summary. The conclusion should tell the reader which was the potential found when IDW, OK and RBF interpolators were used and say clearly the advantage of the use each one for instance.
Author Response
Comments are in the attached document
